# Constraint-Based, Score-Based and Hybrid Algorithms to Construct Bayesian Gene Networks in the Bovine Transcriptome

**DOI:** 10.3390/ani12101305

**Published:** 2022-05-19

**Authors:** Amin Mortazavi, Amir Rashidi, Mostafa Ghaderi-Zefrehei, Parham Moradi, Mohammad Razmkabir, Ikhide G. Imumorin, Sunday O. Peters, Jacqueline Smith

**Affiliations:** 1Department of Animal Science, University of Kurdistan, Sanandaj 66177-15175, Iran; a.mortazavi@agri.uok.ac.ir (A.M.); m.razmkabir@uok.ac.ir (M.R.); 2Department of Animal Science, Yasouj University, Yasouj 74934-75918, Iran; 3Department of Computer Engineering, University of Kurdistan, Sanandaj 66177-15175, Iran; p.moradi@uok.ac.ir; 4School of Biological Sciences, Georgia Institute of Technology, Atlanta, GA 30332, USA; igi2@biology.gatech.edu; 5Department of Animal Science, Berry College, Mount Berry, GA 30149, USA; speters@berry.edu; 6The Roslin Institute, University of Edinburgh, Easter Bush Campus, Midlothian EH25 9RG, UK; jacqueline.smith@roslin.ed.ac.uk

**Keywords:** algorithms, Bayesian network, bovine, endometrium, implantation, DNA microarray

## Abstract

**Simple Summary:**

In this study, we investigated and compared six different Bayesian network algorithms from three different categories to identify hub genes critical to gene expression networks activated in response to progesterone in the bovine uterus. We observed many common hub genes identified between constraint-based algorithms (CBAs) and hybrid algorithms (HAs), while it appeared that score-based algorithm (SBA) methods led to more accurate and relevant predictions of core genes. The results revealed that the identification of hub genes was affected by the type of network reconstruction and by the subsequently used topological parameters. Two identified genes known to have roles during pregnancy are *ISG15* and *DGAT2*. The identified hub genes are associated with biological processes such as amino acid metabolism, hormonal signaling pathways and the immune system. Our analysis revealed a role for miRNAs in the regulation of this system. The biological and physiological roles (enzymatic and hormonal effects) of unannotated identified hub genes should be functionally validated by further studies.

**Abstract:**

Bayesian gene networks are powerful for modelling causal relationships and incorporating prior knowledge for making inferences about relationships. We used three algorithms to construct Bayesian gene networks around genes expressed in the bovine uterus and compared the efficacies of the algorithms. Dataset GSE33030 from the Gene Expression Omnibus (GEO) repository was analyzed using different algorithms for hub gene expression due to the effect of progesterone on bovine endometrial tissue following conception. Six different algorithms (grow-shrink, max-min parent children, tabu search, hill-climbing, max-min hill-climbing and restricted maximum) were compared in three higher categories, including constraint-based, score-based and hybrid algorithms. Gene network parameters were estimated using the bnlearn bundle, which is a Bayesian network structure learning toolbox implemented in R. The results obtained indicated the tabu search algorithm identified the highest degree between genes (390), Markov blankets (25.64), neighborhood sizes (8.76) and branching factors (4.38). The results showed that the highest number of shared hub genes (e.g., proline dehydrogenase 1 (*PRODH*), Sam-pointed domain containing Ets transcription factor (*SPDEF*), monocyte-to-macrophage differentiation associated 2 (*MMD2*), semaphorin 3E (*SEMA3E*), solute carrier family 27 member 6 (*SLC27A6*) and actin gamma 2 (*ACTG2*)) was seen between the hybrid and the constraint-based algorithms, and these genes could be recommended as central to the GSE33030 data series. Functional annotation of the hub genes in uterine tissue during progesterone treatment in the pregnancy period showed that the predicted hub genes were involved in extracellular pathways, lipid and protein metabolism, protein structure and post-translational processes. The identified hub genes obtained by the score-based algorithms had a role in 2-arachidonoylglycerol and enzyme modulation. In conclusion, different algorithms and subsequent topological parameters were used to identify hub genes to better illuminate pathways acting in response to progesterone treatment in the bovine uterus, which should help with our understanding of gene regulatory networks in complex trait expression.

## 1. Introduction

In cattle, fetal death rates from fertilization to term can be up to 56 percent [1]. More than 70 percent of these deaths occur during the first few months of pregnancy [2]. Interestingly, most pregnancy losses occur during the first two to three weeks, when important physiological events, including blastocyst enlongation, pregnancy establishment and embryo implantation, are occurring in the endometrium [1]. Recent studies have shown that the embryo may receive different inputs from the endometrium during its development, including embryotrophic factors (amino acids, carbohydrates, proteins, lipids and other substances) provided by the uterus [1]. Moreover, endometrial gene expression patterns before and after the time of implantation (days 5–20) determine the ability of the uterine environment to maintain pregnancy [3]. Therefore, the identification of genes responsible for the establishment and sustenance of pregnancy in cows could provide key information for the selection of informative genes responsible for increasing bovine fertility [2]. Most reports have found differentially expressed genes in the first two to three weeks of pregnancy, including the insulin-like growth factor (*IGF*) system [1,2,4,5], interleukin 1 (*IL1*) [6], claudin 10 (*CLND10*), matrix Gla protein (MGP) [7], connective tissue growth factor (*CTGF*), solute carrier family 5 member 1 (*SLC5A1*), lactotransferrin (*LTF*) [3,8,9], ubiquitin-like modifier (*ISG15*), complement C1 (*C1*), complement C4 (*C4*), CXC motif chemokine ligand 5 (*CXCL5*), alanyl aminopeptidase (*ANPEP*), fatty acid binding protein 3 (*FABP3*), lipoprotein lipase (*LPL*), solute carrier family 2 member 5 (*SLC2A5*) [3,9], semaphorin 3E (*SEMA3E*), collagen type IV alpha 1 chain (*COL4A1*) and phospholipase A2 (*PLA2*) [2]. It can be inferred that the identification of cause–effect relationships from differentially expressed genes in the uterus within 5 to 16 days of pregnancy may identify biomarkers and core genes responsible for successful and sustained pregnancy.

To date, several methods have been used to extract the network structure of gene expression data. These include linear regression, neural networks, differential equations, Boolean networks and Bayesian networks (BNs) [10,11]. Previous studies have shown that BNs perform better than the other methods for structural reconstruction. BNs are a class of graphical models connecting variables (nodes) by edges (arcs) [10,11]. These networks are examples of the application of graph theory and conditional probability rules for extracting network structures between existing variables of a dataset. BN learning methods are classified into parameter learning and structure learning [12]. Different algorithms have already been proposed for the structure learning of BNs, including constraint-based algorithms (CBAs), score-based algorithms (SBAs), and hybrid algorithms (HAs) [12]. CBA methods seek to estimate the structure of the BN using different conditional independence tests. In CBAs, it is not possible to fully extract the causal relationships between genes; thus, structures reconstructed this way are called partially directed acyclic graphs. CBA methods can be classified into grow-shrink (GS) and max-min parent children (MMPC) algorithms. GS algorithms are used for identifying a Markov blanket (MB) in a BN. MMPC algorithms use a forward-looking selection technique for identifying neighbors in a graph [13]. On the other hand, SBAs are a group of heuristic optimization algorithms that find the best structure according to diverse predefined score functions. In the HA category, two types of algorithms, called maximum-minimum hill-climbing (MMHC) and restricted maximization (RSMAX), have been suggested to solve some limitations of the above-mentioned categories [12]. MMHC uses the features of an MMPC algorithm (to limit search space) and hill-climbing (HC) to find the network with the highest rank in the restricted space [12]. RSMAX is a flexible algorithm that is able to hybridize different characteristics of both the CBA and SBA categories [12].

Given the different characteristics of the aforementioned algorithms but the similar goal (finding the best structure), it is acknowledged that the hub genes identified from each algorithm may be somewhat different. Extracting the important gene networks can help identify genes associated with complex diseases and traits, showing the associations between genes and generalizing the relationships and processes in which genes interact [14]. Hub and regulatory genes in gene networks can be extracted using the above BN categories. Although the best algorithm might be assumed to be the one that hits the most gene annotations, a consensus of all the above BN categories can be taken as the most reliable. In addition, to date, there have not been any comparisons of the effectiveness of different BN algorithms in identifying hub genes in transcriptomic data, especially in cattle.

Therefore, the aims of the present study are to (1) identify hub genes in the bovine uterine transcriptome using two subclasses of the three main algorithms, including CBAs (GS and MMPC), SBAs (HC and TS), and HAs (MMHC and RSMAX); (2) detect hub genes shared between learned structures of the mentioned algorithms; and (3) find the best structural learning BN algorithm for the GSE33030 dataset based on biological classifications and annotations.

## 2. Materials and Methods

### 2.1. Data Used in This Study

The Gene Expression Omnibus (GEO) database under accession number GSE33030 (https://www.ncbi.nlm.nih.gov/geo/query/acc.cgi?acc=GSE33030, (accessed on 2 September 2018) was used for this study. The data were extracted using the GEOquery package [15] in an R environment. The GSE33030 study examined the effect of progesterone on the pattern of bovine endometrium gene expression before and after implantation (days 5 to 16 after fertilization) [8,9]. Five percent (5%) of the genes with the highest variance were considered differentially expressed. The objective of the original project was to ascertain differential effects of elevated P4 concentrations following conception on endometrial gene expression in beef heifers on days 5, 7, 13 and 16 of pregnancy, corresponding to the morula, blastocyst, elongation and maternal recognition pregnancy stages, respectively. Tissues were collected from beef heifers (N = 263) of various treatments on days 5, 7, 13 or 16 of the cycle or pregnancy, and pregnancy was confirmed by the presence of an appropriately developed embryo (conceptus). The total RNA was extracted, and the gene expression was analyzed using bovine Affymetrix microarrays. Differentially expressed genes were selected on the basis of an adjusted *p*-value of <0.01. There were no detectable differences in gene expression in endometria from pregnant and cyclic heifers on days 5, 7 and 13 post-estrus, but the expression of 764 genes was altered due to the presence of the conceptus at the maternal recognition of pregnancy (day 16). Of the genes that were differentially regulated by progesterone, the majority were unique to a specific day of the estrous cycle (early pregnancy). In conclusion, the gene expression in endometria did not differ between pregnant and cycling heifers until day 16 of pregnancy (i.e., the time of maternal recognition of pregnancy and the production of interferon tau by conceptus trophectoderm); however, elevating P4 in early pregnancy caused programmed changes in gene expression in endometria that were hypothesized to impact early conceptus growth and development. Thus, on days 5, 7 and 13, differential gene expression was affected by P4, but on day 16, the conceptus primarily influenced gene expression in the uterine endometria of heifers.

### 2.2. Network Construction

The bnlearn package in an R environment [12] was used to learn the structural BN parameters, such as Markov blanket (MB), neighborhood size (NS) and branching factor (BF). Using a bnlearn toolbox, we used the GS and MMPC algorithms from the CBA category, HC and TS from the SBA category, and MMHC and RSMAX from the HA category for reconstructing the learned structures of the BN. The identification of hub genes was performed by considering the topological degree criterion in the Network Analyzer app of Cytoscape software version 3.6 [16]. Cytoscape is a powerful tool for the graphical visualization of genetic interactions in large databases, as well as the estimation of network topological parameters such as degree, betweenness, closeness and eccentricity. These topological parameters were also estimated for the whole network reconstructed by each algorithm using the Network Analyzer app of Cytoscape software version 3.6 [16]. Some other topological parameters, such as cluster coefficient, shortest path, characteristic path length and neighborhood connectivity (NC), were estimated for structures learned in different algorithms using the igraph package in an R environment [17]. Appendix A shows the pipeline and procedures used in this study.

### 2.3. Downstream Analysis of Hub Genes

The functional annotation information relating to the regulatory hub genes was identified using DAVID [18]. Gene classification, pathway analysis and gene set enrichment analyses of the identified regulatory hub genes were performed using the Panther database [19] with the latest assembly of the *Bos taurus* genome [19]. The results were also examined for gene ontology enrichment and a general comparison of BN structures learned via the different implemented algorithms given. Genes acting as potential upstream regulators of differentially expressed genes were determined using Ingenuity Pathway Analysis (IPA) software (QIAGEN Inc. Redwood City, CA). Over-represented Reactome pathways associated with differentially expressed genes were identified using WebGestalt (WEB-based GEne SeT AnaLysis Toolkit) [http://www.webgestalt.org/, (accessed on 18 May 2021)], [20] with ‘Bos taurus’ as the reference genome and ‘Over-representation analysis’ as the algorithm used.

## 3. Results

### 3.1. Bayesian Structural Network Parameter Estimation

In Table 1, the estimated BN structural parameters using three different algorithmic groups are presented. With SBAs, the highest total numbers of connections and directed connections (388 and 390 for HC and TS, respectively) were observed. With TS from the SBA category, the highest numbers of connections (390), MB (25.64), NS (8.76) and BF (4.38) were estimated. The values for the above-mentioned parameters in the CBA category were higher than those for HAs (89, 2.63, 2 and 0.85 vs. 76, 1.84, 1.76 and 0.85 for total connection, MB, NS and BF for GS and MMHC, respectively).

### 3.2. Topological Parameters of Reconstructed Networks

The topological parameters of the three main BN categories are presented in Table 2. In total, the betweenness values for all six algorithmic groups were in the range of 0.003 (RSMAX) to 0.04 (MMPC), whereas SBAs had the maximum rate of closeness value (0.4 for HC and TS) compared to the two other algorithm categories. The strength values obtained from reconstructed gene networks with the tendency for obtaining separate partitions for SBAs were in the first order, followed by CBAs (2.2 and 3.4 for GS and MMPC, respectively) and HAs (1.7 and 1.5 for MMHC and RSMAX, respectively). Furthermore, the NC values were also maximum for structures obtained with SBAs (9.4 and 9.5 for HC and TS, respectively). The highest characteristic path length values (4.7 and 6 for GS and MMPC, respectively) were observed with CBAs.

### 3.3. Identification of Transcriptomic Hub Genes

In Table 3, we present the bovine uterine transcriptomic hub genes that were identified using three sets of algorithms applied to the GSE33030 DNA microarray dataset. The degree values (D) for each hub gene are also shown for each algorithmic group. The highest degree values were observed in the HC and TS results (about 19). These obtained D values were probably due to the reconstructed causal relationships between all genes in the network learned by SBAs compared to the other two algorithmic groups. The results showed that the first three hub genes were different in each algorithmic category, except for the SBAs. In Table 3, the unique genes that were not common to any of the algorithms are shown in bold. Interestingly, the CBA hub gene degree values were higher than those of HAs (6-4 and 6-5 for GS and MMPC, respectively, vs. 5-3 and 4-3 for MMHC and RSMAX, respectively).

In Figure 1, the transcriptomic hub genes identified by all three algorithmic groups are illustrated. The highest number of common hub genes was identified between HAs–CBAs (6) compared to HAs–SBAs (2) or SBAs–CBAs (1). Most of the identified hub genes in CBAs–HAs were originally from the MMPC–RSMAX pair (*PRODH*, *MMD2*, *SPDEF*, *SEMA3E*, *ACTG2* and *SLC13A5*). Appendix A shows that addressing hub gene detection in the bovine genome has been the core of many studies, but none of them pinpointed hub genes in uterus transcriptomic data.

### 3.4. Downstream Analysis of Identified Genes

#### 3.4.1. Functional Annotation

The results of the functional annotation of identified hub genes from each set of algorithms are presented in Table 4. Only significant pathways (*p* < 0.05) are presented. CBA algorithms highlighted ‘extracellular space’ and ‘alternative splicing’ functions. Genes identified by SBA methods were associated with ‘extracellular space’, ‘secreted proteins’, ‘glycerolipid metabolism’, ‘protease binding’ and ‘disulfide bonds’. The HA algorithms identified hub genes with a role in ‘isopeptide bonds’, ‘alternative splicing’ and ‘arginine and proline metabolism’.

#### 3.4.2. Biological Process

The classifications of biological processes associated with the hub genes identified by CBAs, SBAs and HAs are presented in Figure 2. As shown, in CBAs, cellular process (GO:0009987) (28%) was the most frequent biological process, followed by metabolic process (GO:0008152) (19%) and immune system process (19%) (GO:0002376). Whereas in SBAs, cellular process (20%), response to stimulus (GO:0050896) (17%) and metabolic process (17%) were the first three, ordered respectively. With HAs, similarly to CBAs, cellular process (23%), metabolic process (16%) and immune system process (13%) were the most frequent biological processes associated with the identified hub genes.

#### 3.4.3. Molecular Function

The classifications of molecular functions associated with the hub genes identified by CBAs, SBAs and HAs are presented in Figure 3. Here, all three algorithms produced genes involved in binding (GO:0005488) with a frequency of 46%, 50% and 42% for CBAs, SBAs and HAs, respectively, followed by catalytic activity (GO:0003824), which had the highest frequency in the SBA analysis (40%), followed by the HA (28%) and CBA (26%) analyses, respectively.

#### 3.4.4. Cellular Components

The cellular components associated with the hub genes are illustrated in Figure 4. Interestingly, with SBAs, a different cell component class (extracellular region (GO:0005576) with a frequency of 66%) was observed compared to the cell part (GO:0044464) with frequencies of 44% and 42% from the CBA and HA categories, respectively.

#### 3.4.5. Pathways

The classification of pathways related to hub genes is shown in Figure 5. With CBAs, Huntington’s disease (P00029) (13%) and inflammation mediated by the chemokine and cytokine signaling pathway (P00031) (13%) were the most common pathways shared between the hub genes in this class. A similar phenomenon was observed with HA-derived genes, in which the aforementioned pathways were present with frequencies of 16% and 11%. However, with the SBA analysis, all the identified hub genes were involved in the 2-arachidonoylglycerol biosynthesis (P05726) pathway. Our results suggested that most of the hub genes from the SBA analysis are involved in the synthesis of hormones with steroidal structure (progesterone).

#### 3.4.6. Protein Class

Appendix A shows the classification of the identified hub genes based on their protein class. The results of protein classification showed that, with CBAs, 17% of the hub genes were described as hydrolases (PC00121), followed by cytoskeletal proteins (PC00085) with a frequency of 17%. On the other hand, in the SBA analysis, enzyme modulator (PC00095) and hydrolase (PC00121) were the most significant protein classes, with frequencies of 38% and 15%, respectively. In the analysis using the HA algorithms, four protein classes were represented by a frequency of 15% of all identified hub genes. These protein classes were nucleic acid binding (PC00171), transferase (PC00220), cytoskeletal protein (PC00085) and signaling molecule (PC00207).

### 3.5. Over-Representation Analysis

Differentially expressed genes were analyzed for enriched annotation terms using the WebGestalt suite (http://www.webgestalt.org/, (accessed on 18 May 2021)) [20]. Several Reactome pathways were highlighted. These are shown in Table 5 and include ‘striated muscle contraction’, ‘adipogenesis’, ‘pancreatic secretion’ and ‘white fat cell differentiation’ as being significant (FDR < 0.05).

### 3.6. Ingenuity Pathway Analysis (IPA)

For a more in-depth analysis of the pathways associated with significant network genes, Ingenuity Pathway Analysis software was used (https://www.qiagenbioinformatics.com/products/ingenuitypathway-analysis, (accessed on 20 May 2021).). The top pathways most significantly associated with differentially expressed genes are shown in Figure 6. These included pathways connected with pregnancy hormones (e.g., pregnenolone biosynthesis, estrogen signaling), amino acid transport and metabolism (e.g., g-glutamyl cycle, histidine degradation and serine biosynthesis) and the immune system (e.g., interferon signaling).

An examination was also made of molecules that were likely regulators of identified hub genes. Table 6 shows the most significant molecules that were potential regulators of gene expression within this dataset. These were seen to include progesterone, beta-estradiol, estrogen and 2-methoxyestradiol. Several micro-RNAs were also indicated to potentially play a significant role. These included mir-96, mir-183, miR-182-5p, mi0052-199a-5p and mir-15. mir-199a-5p inhibits the proliferation, movement and angiogenesis of ectopic endometrial mesenchymal stem cells and is known to alleviate endometriosis, while mir-96, mir-182 and mir-183 all help promote cell proliferation, migration and invasion [21,22], and mir-15 is a known tumor suppressor [23].

## 4. Discussion

Using a uterine transcriptomic dataset that examined the effects of progesterone during pregnancy in cattle, we used different modelling algorithms to identify core genes playing significant roles in gene expression networks. In order to create the relevant Bayesian networks (BNs), constraint-based algorithms (CBAs), score-based algorithms (SBAs) and hybrid algorithms (HAs) were all investigated. The SBAs identified the most network connections. The highest total number of connections and directed connections observed with SBAs was probably due to the complete reconstruction of the network structure and the determination of causal relationships between genes identified with these algorithms. Due to their inherent theory, the other two algorithms were not fully capable of determining the causal relationships between all the genes. Theoretically, the concept of the Markov Blanket (MB) refers to the parents, children and spouse of a gene [13]. Extraction of an MB for small-scale networks was one of the network modeling parameters. The aim of the network extraction process by CBAs was to initially identify the MB. Another concept underlying BNs is that of the neighborhood size (NS), which refers to the adjacent genes of a particular gene, plus edges connecting these adjacent nodes, and is very useful in identifying modules in the network. The MB and NS are likely to indicate, in part, the different clusters and nodes in the studied gene expression series. Given the low estimated values for the tabu search (TS) and CBA factors, the probability of the formation of clusters in the GSE33030 dataset was very low. The branching factor (BF), representing the number of genes that can be affected by a specific gene, was one of the factors that created clusters in networks [13]. The RSMAX algorithm from the HAs used a smaller number of conditional independence tests for structural reconstruction compared to the other algorithms.

One of the ways to compare different reconstructed networks is to use their global topological parameters, such as betweenness, eccentricity and degree [14]. Network topology often shows information about the biological importance of a network. Topological parameters help to better recognize the consequences of the hub genes in a network. The degree of connectedness of a gene in a directed graph refers to the number of incoming and outgoing arcs, namely, in-degree and out-degree, respectively. The degrees of genes indicate one of the major topological properties used to identify hub nodes in a graph. We adopted the definition of hub genes as genes with high correlation in the candidate module and high connectivity, as well as required to meet the absolute values of gene module membership (>0.80) and gene trait significance (>0.20) [24]. Interestingly, the reconstructed network obtained by the SBAs showed the highest degree values (7.2 and 7.0 for HC and TS, respectively) compared to the other two algorithms.

The clustering coefficient is a criterion for measuring the tendency of a graph to form consecutive clusters and shows a subset of genes that contains many connections to these genes. The closer the clustering value is to 1, the greater the probability of cluster formation in the gene network [14]. Remarkably, the SBA methods reported the highest clustering coefficient values, showing a relatively high number of co-expressed genes in the network. On the whole among the six different methods, low values of cluster coefficients showed a stochastic structure, as previous studies have shown that biological networks do not show strong tendency to shape clusters [14]. The lowest eccentricity values were obtained in graphs reconstructed by SBAs (3.5 and 3.5 for HC and TS, respectively), indicating high connectivity among genes in the reconstructed gene network. In fact, eccentricity shows the greatest distance between a particular node (‘gene’ in our context) and any other nodes in the graph [14].

The higher number of commonly identified hub genes compared to the other two groups was probably due to common and identical reconstructions by the three main BN algorithmic groups. We recommend using other topological parameters as criteria to identify transcriptomic hub genes. Additionally, the *SEMA3E* hub gene was shared among all three algorithmic groups (HAs, CBAs and SBAs). Research has shown that *ISG15* is a candidate gene for pregnancy recognition or return to the estrus cycle in cows [9]. *ISG15* is also known as a candidate gene for embryo implantation in the uterus. Diacylglycerol O-acyltransferase 2 (*DGAT2*) is another hub gene identified with a biological role in pregnancy. It has eight exons, is found on chromosome 15 and is involved in lipid biosynthesis [25]. This gene encodes one of two enzymes responsible for the catalytic reaction of the final step in triglyceride synthesis in which diacylglycerol attaches to long-chain fatty acyl-CoAs with a covalent bond. It has been shown through previous studies that the protein encoded by this gene is an enzyme involved in the synthesis of milk fat [26] and is known as a marker and candidate gene in determining the fat content of milk [25]. It was also reported that triglyceride is another potential energy source for the bovine blastocyst, and the *DGAT2* catalyst is the final stage in its synthesis [27]. Progesterone appears to stimulate the expression of *DGAT2* in the endometrium [7], and defects of intrauterine growth retardation due to *DGAT2* deficiency were observed, indicating that the presence of this enzyme is necessary for the development of uterine embryos [28]. By regulating the expression of *DGAT2*, progesterone stimulates blastocyst growth in the pre-implantation stage in the uterus [29]. Progesterone injection into the uterus increased the expression of *DGAT2*, which in turn triggered triglyceride synthesis reactions and the transfer of glucose in the uterus [3]. Increasing the expression of *DGAT2* through progesterone led to the secretion of histotroph via the endometrium. *DGAT2* was recognized as a hub gene in our CBA analysis. However, other identified hub genes with no clear biological roles may be involved in metabolic or immune processes. The results in Table 3 show that identification of hub genes by each algorithmic category was partly affected by the reconstructed graph, as well as by the implemented topological parameters.

A useful tool for investigating reproductive problems in livestock could be to examine the expression of these core network genes in different tissues. In this way, general hubs and tissue-specific hubs can be identified. In order to find general hub genes ranked high in a set of tissues, the rank product method could be used [30]. We believe that MB can be used to mine the whole network for ranked genes by the number of neighbors in the gene network. A gene’s rank product is the product of its ranks from each network. For locating hubs specific to a group of tissues, rank product could be used to rank hubs in both the target group of tissues and all other tissues, separately.

The hub genes derived from CBA methodology were involved in extracellular and alternative splicing pathways, with the annotation of hub genes from SBA analysis showing genes involved in extracellular pathways, secretion, glycerolipid metabolism and the formation of disulfide bonds [31]. The annotation of the hub genes derived from the HA analysis also included genes involved in extracellular pathways, alternative splicing and the metabolism of arginine and proline [31]. Based on the original GSE33030 dataset, high concentrations of progesterone secretion during the period of 7 to 16 days of pregnancy were conducive for embryo implantation in the uterus, indicating that during this stage, the pathways and genes relating to the production of hormones, enzymes and molecules related to the preservation and continuation of pregnancy were heavily activated. The production of growth hormones, binding molecules, chemokines and cytokines ensures the uterus is ready for embryo implantation [32,33]. The results of the hub gene annotation in this study also indicated that alternative splicing and extracellular pathways are important. The highest numbers of identified hub genes (7 to 8) that were identified by all the algorithms were in these pathways. Therefore, the pathways for the synthesis of hormones and enzymes were essential for preparing the uterus for implantation during days 5 to 13 of pregnancy. In cell biology, the extracellular space refers to gene products that exit from the plasmid membrane and flow through the intercellular fluids. Extracellular compounds include metabolites, ions, proteins and products such as RNA, DNA, lipids and microbial products, which affect endometrium function. Alternative splicing also plays an important role in the diversity of proteins derived from a particular transcript.

In the SBA analysis, the pathways for protease binding, disulfide bonds and glycerolipid metabolism were also significant. Isopeptide bonds also play a role in the binding of two amino acids forming polypeptides. Disulfide bonds interconnect between polypeptide units in proteins and form the tertiary structure of proteins, whereas the “protease bonds” category of genes are involved in protein decomposition. The metabolism of glycolipids is also essential for the synthesis of progesterone steroid hormones and other genes (*DGAT2*) in preparing the uterus for implantation (days 5 to 13). The hub gene annotation results showed that genes identified using SBAs and HAs were enriched for binding pathways (disulfide bands, isopeptide and protease), the production of enzymes and protein products (extracellular space and alternative splicing) and the synthesis of progesterone steroid hormone (glycerolipid metabolism). Therefore, considering the hub gene annotation results led to a more accurate and relevant prediction of the genes involved in the pathways of enzyme synthesis, the binding of polypeptide units and the synthesis of progesterone-related hormone during pregnancy establishment. In comparison to CBAs and HAs, the SBA methods found genes more related to the experimental treatment (in this case, the progesterone effect from days 5 to 16 of pregnancy in the GSE33030 microarray study). However, it is important to investigate the relevant biological and physiological functions of genes identified by all three algorithmic groups. These results classified hub genes based on their biological processes, molecular functions, cellular components, pathways and protein classes for further investigation. It was shown that models of gene regulation differed depending on the biological state of dairy cattle. Therefore, the importance of hub genes should be determined within the relevant biological context.

Cellular processes involve complicated cascades of biochemical reactions and signaling pathways. For correct cell function, these processes are required to be tightly controlled. Dysregulation of any element of these pathways can lead to a vast array of pathologies. By elevating progesterone during days 7 to 12 of estrus or pregnancy, a series of processes to synthesize and secrete progesterone for embryo implantation are activated, leading to the enriched gene expression of metabolic processes. During preimplantation, endometrial gene expression is regulated by the secretion of progesterone and interferon tau, and patterns of endometrium gene expression may be regulated only by progesterone and interferon, or by both [34]. Embryo implantation is also seen as an inflammatory immune response [6]. Interferon tau is a type I interferon that plays an antiviral, anticoagulant and immune-stimulating role. Interferon tau induces the expression of a number of genes in the endometrium that are essential for the transfer of food to the embryo or increase the expression of genes necessary to prepare the endometrium for implantation and continued pregnancy. Most of the hub genes identified from the CBA and HA analyses participated in immune system processes.

The results of this study were based on the use of bovine Affymetrix microarrays. The biological validation of the identified genes may be influenced by data type with respect to using microarrays versus RNA-Seq [6]. Although a limitation of this study was the lack of another independent dataset in the database for validation of the hub genes, the goal of comparing three common algorithms for the detection of hub genes shared between learned structures of the algorithms was achieved. RNA-Seq allows full sequencing of the whole transcriptome, while microarrays only profile predefined transcripts and genes through hybridization. The ability of RNA-Seq to identify more differentially modulated transcripts of biological relevance, splice variants and non-coding transcripts, such as microRNAs, long non-coding RNAs and pseudogenes, makes it superior to microarrays. This difference has additional implications for mechanistic investigations or biomarker discovery [6] making RNA-Seq data more useful with higher predictive power [6].

## 5. Conclusions

In this study, we compared the ability of six different BN algorithms from three different categories (CBAs, SBAs and HAs) to identify hub genes critical to gene expression networks activated in response to progesterone in the bovine uterus. We observed many common hub genes identified between the CBAs and HAs, while it appeared that SBA methods led to more accurate and relevant predictions of core genes. The results of this study revealed that the identification of hub genes was affected by the type of network reconstruction and by the subsequently used topological parameters. *ISG15* and *DGAT2* are two identified genes known to have roles during pregnancy. Other hub genes are associated with biological processes such as amino acid metabolism, hormonal signaling pathways and the immune system. Our analysis revealed a role for miRNAs in the regulation of this system. The biological and physiological roles (enzymatic and hormonal effects) of unannotated identified hub genes should be functionally validated by further studies.

## Figures and Tables

**Figure 1 animals-12-01305-f001:**
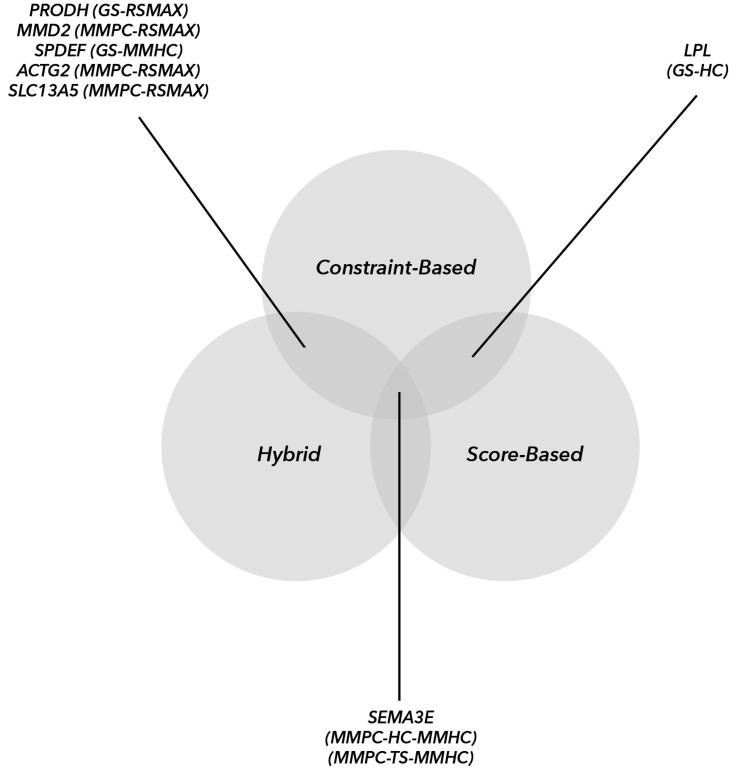
Venn diagram of commonly identified hub genes from all three algorithmic groups. The parentheses represent the subclasses of the overlapped algorithms. GS—grow-shrink; MMPC—max-min parent children; HC—hill-climbing; TS—tabu search; MMHC—max-min hill-climbing; RSMAX—restricted maximum.

**Figure 2 animals-12-01305-f002:**
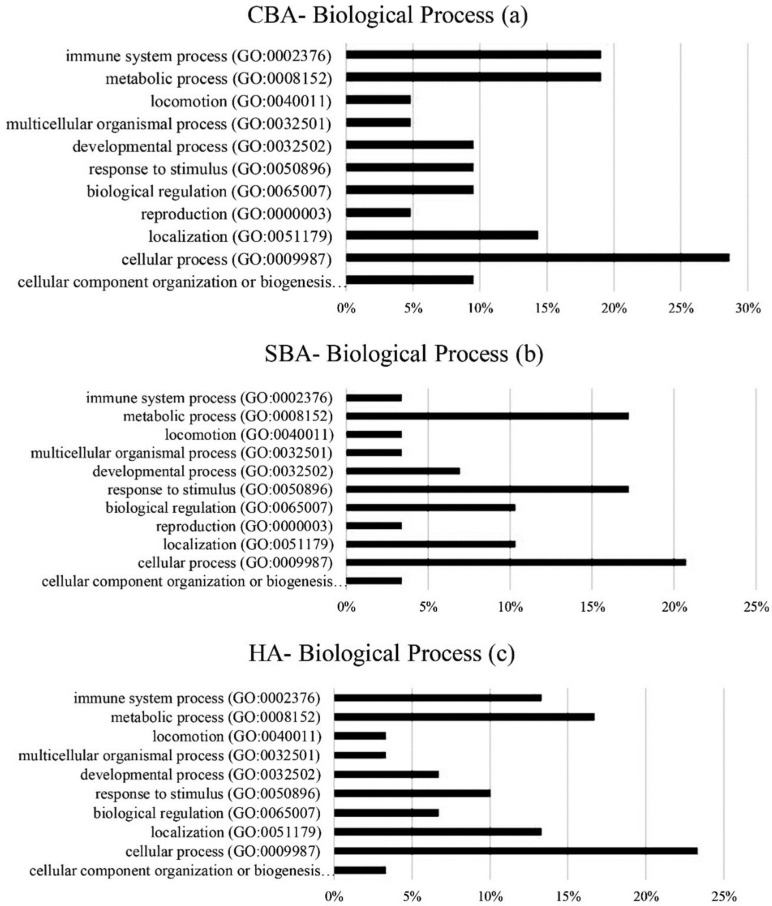
Biological processes of identified hub genes from CBAs (**a**), SBAs (**b**) and HAs (**c**). CBAs—constraint-based algorithms; SBAs—score-based algorithms; HAs—hybrid algorithms.

**Figure 3 animals-12-01305-f003:**
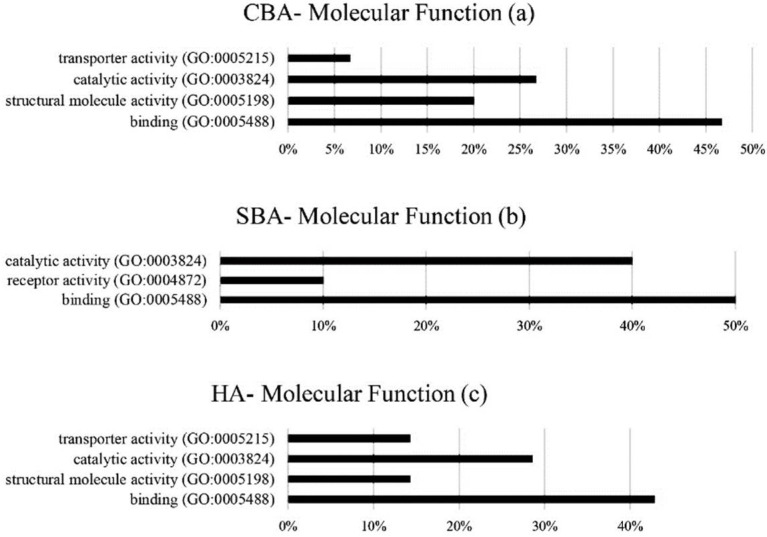
Molecular functions of identified hub genes from CBAs (**a**), SBAs (**b**) and HAs (**c**). CBAs— constraint-based algorithms; SBAs—score-based algorithms; HAs—hybrid algorithms.

**Figure 4 animals-12-01305-f004:**
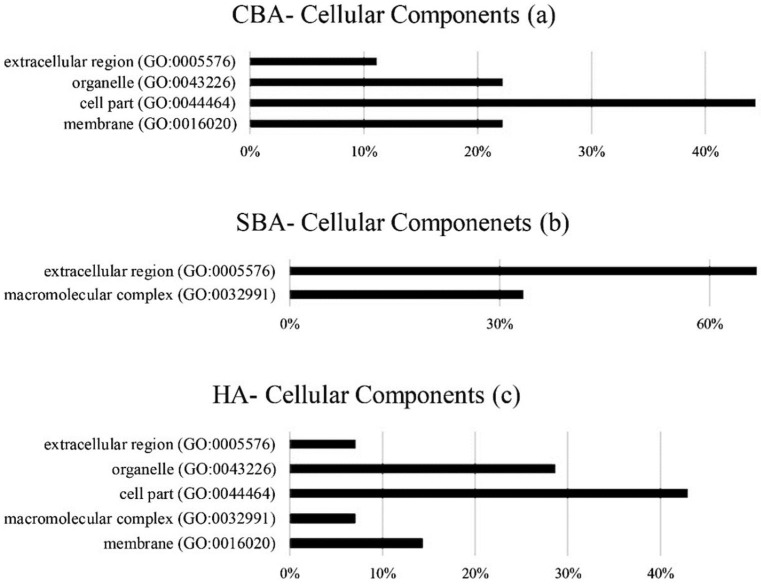
Cellular components of identified hub genes from CBAs (**a**), SBAs (**b**) and HAs (**c**). CBAs—constraint-based algorithms; SBAs—score-based algorithms; HAs—hybrid algorithms.

**Figure 5 animals-12-01305-f005:**
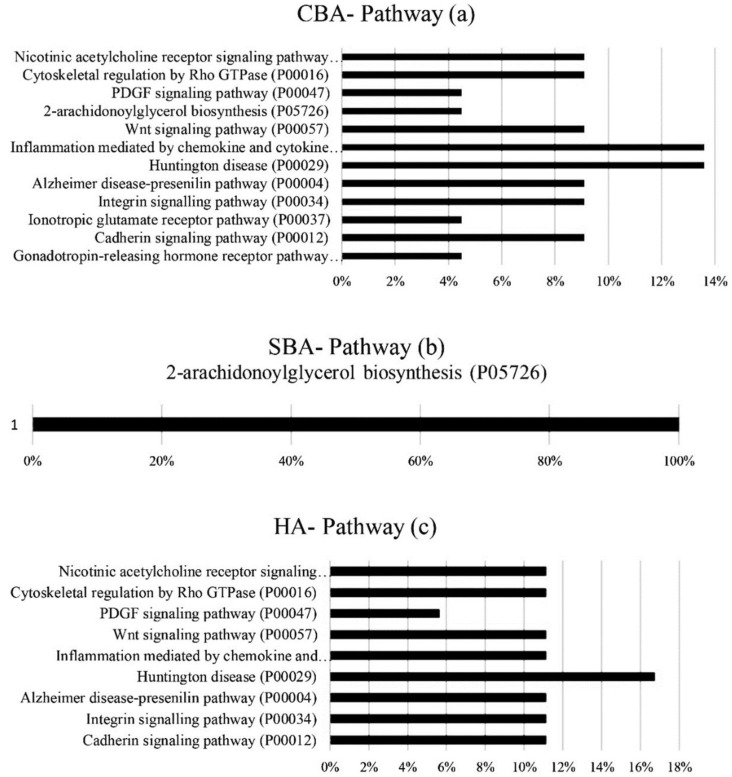
Pathways associated with hub genes from (**a**) CBA, (**b**) SBA and (**c**) HA algorithms. CBAs—constraint-based algorithms; SBAs—score-based algorithms; HAs—hybrid algorithms.

**Figure 6 animals-12-01305-f006:**
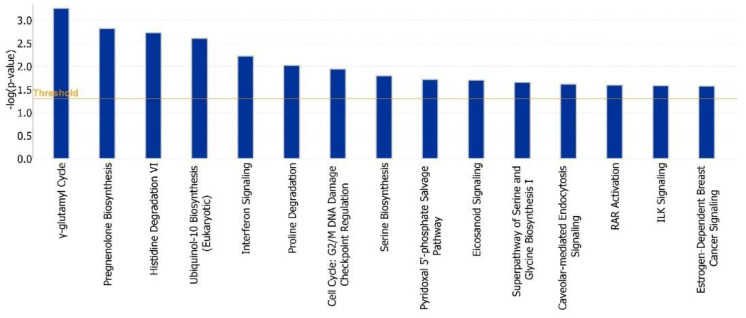
Biological pathways identified from Ingenuity Pathway Analysis (IPA) of hub genes.

**Table 1 animals-12-01305-t001:** Estimation of structural Bayesian network parameters using different algorithms.

Parameters	CBA		SBA		HA	
	GS	MMPC	HC	TS	MMHC	RSMAX
No. of nodes	89	89	89	89	89	89
No. of arcs (edges)	89	77	388	390	76	70
Undirected edges	13	0	0	0	0	0
Directed arcs	76	77	388	390	76	70
MB	2.63	1.73	25.3	25.6	1.8	1.6
NS	2	1.73	8.7	8.7	1.7	1.5
BF	0.85	0	4.3	4.3	0.85	0.79
No. of Tests	43,903	55,335	43,338	43,338	55,601	23,662

Abbreviations: CBA: constraint-based algorithm; SBA: score-based algorithm; HA: hybrid algorithm; GS: grow-shrink; MMPC: max-min parent children; HC: hill-climbing; TS: tabu search; MMHC: max-min hill-climbing; RSMAX: restricted maximize; MB: Markov blanket; NS: neighborhood size; BF: branching factor.

**Table 2 animals-12-01305-t002:** Topological parameters of Bayesian network by different implemented algorithms.

Parameters	CBA		SBA		HA	
	GS	MMPC	HC	TS	MMHC	RSMAX
Betweenness	0.008507684	0.04031794	0.009298721	0.009213601	0.001715626	0.0003272751
Eccentricity	10.95506	10.48315	3.505618	3.505618	10.48315	4.033708
Degree	1.93258427	2.674157	7.228571	7.011236	1.348315	1.191011
Closeness	0.04408607	0.02552964	0.4348416	0.4358546	0.0255294	0.01263703
Centrality	3.404494	6.438202	4470240	5490951	6.438202	3.404494
Strength	2.292135	3.460674	8.719101	8.764045	1.707865	1.573034
CC	0.005	0.019	0.047	0.048	0	0
SP	685 (8%)	1777 (22%)	4736 (60%)	4736 (60%)	256 (3%)	107 (1%)
CPL	4.749	6.023	3.322	3.319	3.636	1.748
NC	2.884684685	3.529701	9.47985	9.550446	2.318357	2.191799

Abbreviations: CBA: constraint-based algorithm; SBA: score-based algorithm; HA: hybrid algorithm; GS: grow-shrink; MMPC: max-min parent children; HC: hill-climbing; TS: tabu search; MMHC: max-min hill-climbing; RSMAX: restricted maximize; CC: cluster coefficient; SP: shortest path; CPL: characteristic path length; NC: neighborhood connectivity.

**Table 3 animals-12-01305-t003:** Hub genes identified by the various Bayesian network algorithms used in this study.

CBA				SBA				HA			
GS	D	MMPC	D	HC	D	TS	D	MMHC	D	RSMAX	D
LPL	6	SEMA3E	6	CSTB	19	CSTB	19	BOLA	5	ISG15	4
PRODH	5	ACTG2	6	DGKI	16	DGKI	16	SPDEF	4	PRODH	4
**GNMT**	5	BOLA	6	SEMA3E	16	SEMA3E	16	**FOXRED2**	4	MMD2	4
BOLA	4	SLC13A5	6	DAPL1	15	DAPL1	15	SEMA3E	4	**SLC27A6**	3
**UPK1B**	4	SPDEF	6	**DPP4**	13	LPL	13	ACTG2	4	ACTG2	3
**ANPEP**	4	MMD2	6	LPL	13	DKK1	13	ISG15	4	**CKMT1**	3
SPDEF	4	**MICAL2**	5	**DKK1**	13	LOC783399	13	SLC13A5	3	**DEPDC5**	3
**CAMK2B**	4	**DGAT2**	5	LOC783399	13	**BoLA-DRB3**	13	**CHEK1**	3	BOLA	3

Abbreviations: CBA: constraint-based algorithm; SBA: score-based algorithm; HA: hybrid algorithm; GS: grow-shrink; D: degree criterion; MMPC: max-min parent children; HC: hill-climbing; TS: tabu search; MMHC: max-min hill-climbing; RSMAX: restricted maximum. The genes in bolded format represent unique genes among all the algorithmic categories.

**Table 4 animals-12-01305-t004:** Functional annotation of identified hub genes.

Term	Count	*p*-Value	FE	Bonferroni	Benjamini	FDR
CBA						
Extracellular space	4	0.04	3.2	0.8	0.8	48
Alternative splicing	8	0.009	1.9	0.3	0.3	8
**SBA**						
Extracellular space	4	0.01	7.7	0.22	0.2	5
Secreted	4	0.01	5.9	0.50	0.5	12
Glycerolipid metabolism	2	0.03	47.6	0.28	0.2	18
Protease binding	2	0.04	47.7	0.68	0.6	26
Disulfide bond	4	0.04	3.9	0.77	0.7	32
**HA**						
Isopeptide bond	3	0.05	6.8	0.93	0.9	40
Alternative splicing	7	0.02	1.7	0.97	0.8	51
Arginine and proline metabolism	2	0.0	1.7	0.1	0.1	9

Abbreviations: FE—fold enrichment; FDR—false discovery rate; CBA—constraint-based algorithm; SBA—score-based algorithm; HA—hybrid algorithm.

**Table 5 animals-12-01305-t005:** Over-represented Reactome pathways.

Gene Set	Description	Size	Expect	Ratio	*p* Value	FDR
WP216	Striated muscle contraction	45	1.33	12.782	3.22 × 10^−15^	1.63 × 10^−12^
WP447	Adipogenesis genes	134	3.9604	4.2924	3.84 × 10^−7^	9.7015 × 10^−5^
mmu04972	Pancreatic secretion	103	3.0442	3.6134	0.00022	0.037012
WP2872	White fat cell differentiation	32	0.94578	6.344	0.000299	0.037763
WP4344	Sphingolipid metabolism (general overview)	25	0.73889	6.7669	0.000711	0.059801
WP512	Id signaling pathway	51	1.5073	4.644	0.000693	0.059801
WP4690	Sphingolipid metabolism (integrated pathway)	26	0.76845	6.5066	0.000859	0.061947
mmu00600	Sphingolipid metabolism	48	1.4187	4.2293	0.002729	0.17224
WP2084	SREBF and miR33 in cholesterol and lipid homeostasis	11	0.32511	9.2276	0.003533	0.19822
WP1596	Iron homeostasis	15	0.44333	6.7669	0.008926	0.38976

**Table 6 animals-12-01305-t006:** Predicted upstream regulators of differentially expressed genes in this study.

Upstream Regulator	Molecule Type	*p*-Value of Overlap	Target Molecules in Dataset
progesterone	chemical—endogenous mammalian	8.44 × 10^−10^	ADAMDEC1, CFTR, CYP26A1, DKK1, DPP4, EDN3, GNMT, IGFBP1, LPL, LTF, NPL, PDZK1IP1, PRSS35, PTGS2, SFRP4, STAT5A
JAK	group	6.32 × 10^−9^	FBXO32, IFIT1, ISG15, PTGS2, RSAD2, STAT5A
STAT3	transcription regulator	2.38 × 10^−8^	CHEK1, CYP26A1, DKK1, DPP4, HLA-DQA1, IFIT1, IGFBP1, ISG15, LTF, MAP2, PTGS2, RSAD2, TFF3, TRPM3
ACOX1	enzyme	2.95 × 10^−8^	CSTB, CYP26A1, DPP4, GNMT, HLA-DQA1, IGFBP1, LPL, UCP2
mir-96	microRNA	0.000000319	IFIT1, IGFBP1, ISG15, RSAD2, SAMD9
dexamethasone	chemical drug	0.000000765	ACTG2, ANPEP, CHEK1, CHGA, CYP26A1, DGAT2, EDN3, FBXO32, GGT1, IFIT1, IGFBP1, ISG15, ITGB5, LOC102724788/PRODH, LPL, MAP2, MSTN, OR51E1, PTGS2, RSAD2, STAT5A, TOP2A, UCP2
mir-183	microRNA	0.000000841	IFIT1, IGFBP1, ISG15, RSAD2, SAMD9
miR-182-5p (and other miRNAs w/seed UUGGCAA)	mature microRNA	0.00000239	IFIT1, IGFBP1, ISG15, RSAD2, SAMD9
beta-estradiol	chemical—endogenous mammalian	0.0000033	ADAMDEC1, ANPEP, BLOC1S6, CFTR, CHEK1, CHGA, CSTB, DKK1, HLA-DQA1, IGFBP1, ISG15, LPL, LTF, MEDAG, PDZK1IP1, PRSS35, PTGS2, SFRP4, SLC6A20, SPDEF, STAT5A, TFF3, TOP2A
miR-199a-5p (and other miRNAs w/seed CCAGUGU)	mature microRNA	0.00000526	ACTG2, CSTB, ISG15, PTGS2, RSAD2
estrogen	chemical drug	0.00000567	CFTR, CRABP1, IGFBP1, LPL, LTF, PTGS2, STAT5A, TOP2A
mir-15	microRNA	0.00000867	CHEK1, IFIT1, ISG15, PTGS2, UCP2
ciprofibrate	chemical drug	0.00000957	CSTB, CYP26A1, DPP4, GNMT, IGFBP1, LPL
GALNT6	Enzyme	0.0000127	ANPEP, DPP4, TFF3
2-methoxyestradiol	chemical—endogenous mammalian	0.0000155	ITGB5, LOC102724788/PRODH, LTF, PTGS2

## Data Availability

Data supporting reported results are contained within the article. All datasets collected and analyzed during the current study are available from the corresponding author on reasonable request.

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
