# Peer review of "Constraint-Based, Score-Based and Hybrid Algorithms to Construct Bayesian Gene Networks in the Bovine Transcriptome"

_animals, 2022, doi:10.3390/ani12101305_

Round 1
Reviewer 1 Report
Authors explained their objectives and improved their manuscript.
English has to be checked carefully.
Definition of hub genes should be given in terms of connectivity, the absolute value of the gene module membership and gene-trait-significance.
Validation of hub genes in a second independent data set should be discussed. Conclusions on hub genes may be too strong due to missing validation.
Author Response
Response to Reviewers Comments
Reviewer Comment
Definitions of hub genes should be given in terms of connectivity, the absolute value of gene module membership and gene-trait significance
Response
Authors have included the definition of hub genes as stated by the reviewer above in the discussion section of the paper. The reviewer is invited to look at lines 338 – 340 of the revised manuscript.
Comment
Validation of hub genes in a second independent data set should be discussed. Conclusions of hub genes may be too strong due to missing validation.
Response
We have included a statement in lines 452 -455 in the revised manuscript that indicated that we were not able to find another independent dataset that is comparable to the one we used (progesterone in the bovine uterus) A limitation of this study is the lack of another independent data set in the database for validation of the hub genes, the goal of comparing three common algorithms for detection of hub genes shared between learned structures of the algorithms was achieved.

Reviewer 2 Report
No furhter comments.
Author Response
No comment from reviewer 2
